# Flake ZnAl Alloy as an Effective Pigment in Silicate Coatings for the Corrosion Protection of Steel

Nguyen Hoang [1,*], Truong Anh Khoa [1], Le Thi Nhung [1], Phan Minh Phuong [1], Tran Dinh Binh [1], To Thi Xuan Hang [2], Nguyen Van Chi [3] and Thanh-Danh Nguyen [4]

1 Nha Trang Institute of Technology Research & Application, Vietnam Academy of Science and Technology, Nha Trang 58000, Vietnam; truonganhkhoa@nitra.vast.vn (T.A.K.); lethinhung@nitra.vast.vn (L.T.N.); minhphuong345@yahoo.com.vn (P.M.P.); trandinhbinh51@gmail.com (T.D.B.)

2 Institute for Tropical Technology, Vietnam Academy of Science and Technology, 18 Hoang Quoc Viet, Cau Giay, Hanoi 1000000, Vietnam; ttxhang60@gmail.com

3 Coastal Branch, Vietnam—Russia Tropical Center, 30 Nguyen Thien Thuat, Nha Trang 58000, Vietnam; nguyenvanchirvtc@gmail.com

4 Institute of Chemical Technology, Vietnam Academy of Science and Technology, 1A, TL29, Thanh Loc Ward, District 12, Ho Chi Minh City 70000, Vietnam; ntdanh@ict.vast.vn

* Correspondence: nguyenhoangnt77@nitra.vast.vn

**Abstract:** Spherical zinc is well known as an effective pigment for the corrosion protection of carbon steel. However, a high proportion of spherical Zn in a coating leads to difficulties in fabricating the coating solution and increased cost. In this work, the influence of flake ZnAl alloy in silicate coatings on the corrosion protection properties of steel substrates was investigated. The electrochemical behaviour of coatings containing different flake ZnAl alloy content immersed in NaCl solution (3.5 wt%) was evaluated using an electrochemical impedance spectroscopy (EIS) method. A salt spray test was performed to evaluate corrosion on the coating surface. Pull-off tests of the coatings before and after the salt spray process were performed, and the surface morphology was analysed to determine the degradation of corrosion resistance. The results show that silicate coating containing flake ZnAl alloy (25 wt%) possesses the highest total resistance (1417 Ω) and the longest time to the appearance of white rust (720 h). The surface morphology of the coating containing 25 wt% flake ZnAl alloy was found to include corrosion products with the most compacted surface, which effectively prevents the penetration of electrolytes to the interface between the coating and the steel.

**Keywords:** flake ZnAl pigment; corrosion protection; steel; inorganic coating; silicate coating

## 1. Introduction

Surface coating using metals is an effective method for protecting carbon steel against corrosion. Zn-rich coatings are widely used and are considered to be effective as protective coatings for steel surfaces because of their negative standard corrosion potential [1]. Zn-rich coatings may be organic or inorganic [2]. Organic Zn-rich coatings include volatile organic compounds (VOCs), e.g., benzene and ethanol, which are harmful to human health and the environment, whereas inorganic coatings often use nontoxic water-based solvents [3]. Thus, researchers aim to develop water-based Zn-rich coatings for environmentally friendly paints.

Currently, spherical Zn is used in the application of water-based inorganic Zn-rich coatings for the corrosion protection of carbon steel. To ensure that a coating has the required electrochemical protection properties, the level of Zn must be sufficient, usually at least 80% [2]. However, with a high concentration of spherical Zn, the coating solution is heterogeneous due to a low ratio of surface area to weight. As a result, such a solution can contain serious defects in the coating [4]. To overcome this drawback, flake zinc powder has been used as an effective alternative. Kalendova et al. [5] showed that the presence of flake

Zn pigment in an organic coating could improve corrosion protection in comparison to spherical Zn due to its superior properties. More importantly, flake Zn has a higher surface-area-to-weight ratio, leading to better electrical contact between Zn particles and lower current density in the Zn-rich coating due to good suspension in the paint solution [6–8]. Furthermore, the low quantity of flake Zn in the coating can significantly reduce the cost.

Although Zn-rich coating provides effective electrochemical protection of the base metals, the stability of Zn, a strongly chemically active metal, is not high in aggressive environments. In order to enhance the corrosion resistance of Zn-rich coatings, many studies have used film-forming or pigment-modifying additives. Combination of Al and Zn pigments in the coating could significantly improve the corrosion protection of Zn-rich coatings [9,10]. Zn-rich coatings with Al content exhibit better cathodic protection on steel substrates, preventing oxidative processes of zinc particles [11–13]. In addition, silicate coatings containing Al powder not only provide a cathodic protection property but also react with alkali metal silicates to produce insoluble silicates and form a thick film [14].

The use of ZnAl alloy pigment can increase the coating stability due to the passivation of Al, and the colour of this pigment is also more distinctive. On the other hand, due to the ductile characteristic of the ZnAl alloy, its flake powder is easily fabricated in nanometre thickness, which can improve the dispersibility and the corrosion protection and reduce the required amount of zinc [15,16].

In our recent study [17], the corrosion protection of carbon steel was found to be significantly improved when some of the spherical Zn was replaced by a combination of Zr conversion and flake ZnAl alloys in the silicate coatings. To the best of our knowledge, the effect of flake ZnAl alloy pigment in a silicate coating on the corrosion protection of metals has not been studied. In this work, the influence of ZnAl alloy content on the corrosion protection properties of silicate coatings on steel substrates was evaluated. Silicate coatings containing different concentrations of flake ZnAl alloy were explored, and the anticorrosive properties and morphology were evaluated to demonstrate the performance of the protective coatings.

## 2. Materials and Methods

### 2.1. Material and Sample Preparation

An aqueous potassium silicate solution was obtained from Xingtai Ocean Chemical Company. A 30 wt% nano-silica solution (particle size range of 9–10 nm) was provided by Vietnam Investment Casting Ltd. Co and was added into the potassium silicate solution to produce a binder ($SiO_2:K_2O$ = 5, mol/mol) in an inorganic coating. Flake Zn-Al alloy (size of 5–7 μm, Zn:Al = 80:20) was purchased from Hunan Jinhao New Material Technology Co. (Changsha, China), and two additives, Silquest A187 (Momentive Performance Materials Inc., New York, NY, USA) as a dispersing agent and tributyl phosphate (Xilong Scientific Co., Santou, China) as an antifoaming agent, were used.

### 2.2. Preparation of Coatings

Carbon steel samples with a size of 10 cm × 15 cm × 0.2 cm were used as metallic substrates, ground with 600–1200 grade SiC papers, and then degreased in methanol solutions under ultrasonic irradiation. The steel samples were rinsed with the water and dried at room temperature. Silicate solution was produced from potassium silicate solution with a binder, additives, and various pigments to form the silicate coatings with different coating formulations of flake ZnAl alloys (FZnAl20 (20 wt%); FZnAl25 (25 wt%); FZnAl30 (30 wt%)). The carbon steel samples were prepared using a spray-coating method with a coating thickness (100 ± 10 μm) determined after drying at room temperature for 7 days.

*2.3. Methods*

2.3.1. Electrochemical Measurements

Electrochemical investigation was carried out on coated and bare samples using a three-electrode cell device (Autolab PGSTAT 204N, Ionenstrasse, Herisau, Switzerland). The system was assembled with the auxiliary platinum electrode, the test sample used as the working electrode, and a reference Ag/AgCl electrode. DC polarization was used to measure the parameters of the coated samples, including corrosion current density ($i_{\text{corr}}$) and corrosion potential ($E_{\text{corr}}$). Polarization resistance ($R_P$) was estimated via the Stern–Geary Equation (1) [16]:

$$R_p = \frac{\beta_a \beta_c}{2.303(\beta_a + \beta_c) i_{\text{corr}}} \tag{1}$$

where $\beta_a$ and $\beta_c$ are anodic and cathodic Tafel constants (V/decade), respectively.

The voltages were scanned at $\pm 150$ mV with respect to open-circuit potential (OCP) at a scan rate of 0.01 V/s. The $i_{\text{corr}}$ values were obtained using Tafel extrapolation at $\pm 50$ mV with respect to the OCP. Electrochemical impedance spectroscopy (EIS) investigation was performed on an area of 3.46 cm$^2$ for the silicate coatings in NaCl solution (3.5%, $w/w$) at the frequency range from 100 kHz to 0.01 Hz. The amplitude of the alternating potentials was applied in the range from 10 mV to 0 with respect to OCP. The tests were performed at least three times for each sample, and the data were analysed by Nova 2.0 software.

2.3.2. Pull-Off Test

The pull-off adhesion strength of the silicate coatings was assessed on a Defelsko Positest AT (ASTM-D4541, DeFelsko Co., New York, NY, USA). Dollies (diameter of 20 mm) were fixed to the coated samples by a two-part Araldite 2015 adhesive (Huntsman Co., osnabrück, Germany). The sample was dried under room temperature until the glue was fully cured (about 24 h). Then, a slot around the dollies was made and pulled at a speed of 10 mm min$^{-1}$ on the coating surface until the coatings were detached from the substrates. All tests were performed at least three times. The decrease in adhesion strength was calculated using Equation (2):

$$\text{Adhesion loss } (\%) = \frac{\text{Adhesion before salt spray test} - \text{adhesion after salt spray test}}{\text{Adhesion before test}} \times 100\% \tag{2}$$

2.3.3. Salt Spray Test

Salt spray tests were evaluated in a salt spray box (Q-FOG CCT600, Q-lab Co., Cleveland, OH, USA) according to JIS 8502:1999. The rest of the coating surface was protected using a waterproofing mixture of beeswax–colophony (3:1). All tests were performed at least three times. Elemental composition and morphology analysis of the samples was carried out using field emission scanning electron microscope (FESEM, JEOL, Tokyo, Japan) combined with energy-dispersive X-ray spectroscopy (EDS) (Jeol 6490 JED 2300).

**3. Results and Discussion**

Zn-rich inorganic coatings have good electrical conductivity, weather resistance, and solvent resistance properties [18]. However, due to the high Zn content in the coating, these coatings have many disadvantages, including surface voids, poor insulation performance, and brittleness [19–21]. To reduce the Zn content in the coating, a flake ZnAl alloy pigment was investigated. Furthermore, wetting agents were added to the pigment powder for improvement of the dispersion and the ability to grind the pigment powder to prepare the flake ZnAl alloy [1], with a low critical wetting surface tension of the lamellar pigments. The flake ZnAl alloy content should be less than 30 wt% in the silicate coating to ensure dispersion and uniformity. The anticorrosion behaviour and adhesion of the coatings with different ZnAl alloy contents were studied via EIS measurement and salt spray test. SEM

images were analysed to investigate the morphology of the coatings before and after the salt spray test.

### 3.1. Corrosion Rate of Silicate Coating

To study the corrosion resistance of the silicate coatings containing different pigments contents, measurement of the polarization curves was performed with the samples immersed in NaCl solution (3.5 wt%), as shown in Figure 1. The polarization parameters extrapolated from the Tafel chart are given in Table 1.

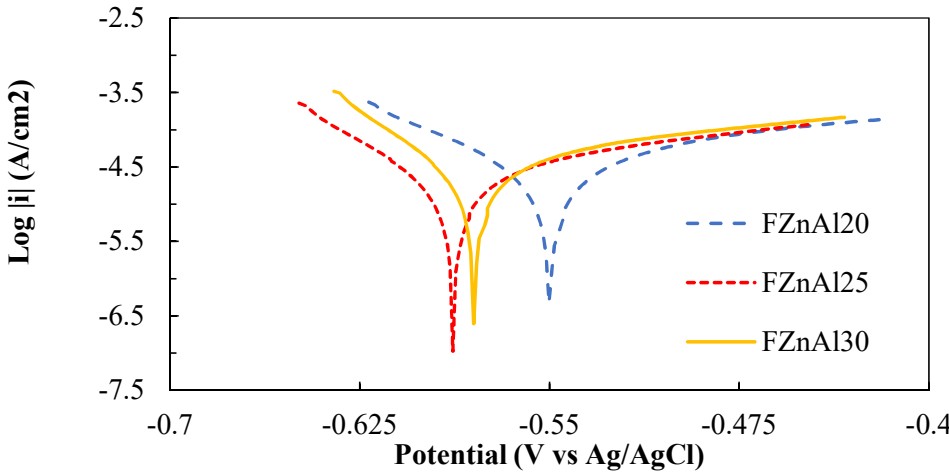

**Figure 1.** DC polarization curves for silicate coating containing different ZnAl alloy concentrations after immersion in NaCl (3.5 wt%) for 48 h.

**Table 1.** Data obtained from DC polarization curves for silicate coating containing different ZnAl alloy concentrations after immersion in NaCl (3.5 wt%) for 48 h.

| Samples | $E_{corr}$ (mV) | $i_{corr}$ ($\mu$A/cm$^2$) | $\beta_a$ (mV/dec) | $-\beta_c$ (mV/dec) | $R_P$ ($\Omega.$cm$^2$) |
|---|---|---|---|---|---|
| FZnAl20 | −549.41 | 12.77 | 105.52 | 192.99 | 2319.30 |
| FZnAl25 | −587.53 | 7.48 | 69.73 | 160.99 | 2823.67 |
| FZnAl30 | −579.33 | 9.65 | 60.35 | 160.69 | 1974.55 |

The results show that the potential of the samples coated with FZnAl20, FZnAl25, and FZnAl30 strongly shifts to more positive positions than the reference potential of Ag/AgCl (−735 mV), demonstrating that Zn-rich coatings act primarily as cathodic protection mechanisms [22]. It indicates that the flake ZnAl pigments do not protect the steel substrate against corrosion by a sacrificial anodic mechanism but mainly by a barrier mechanism. As can be seen in Table 1, the effect of the flake ZnAl pigment content of the silicate coating is evident in both parameters $i_{corr}$ (7.48 to 12.77 $\mu$A/cm$^2$) and $E_{corr}$ (−549.41 to −587.53 mV). The lowest value of $i_{corr}$ is found in the samples coated with the FZnAl25 pigment. Thus, the appropriate content of the flake ZnAl pigment can significantly improve the barrier properties by filling the pores on the surface, thereby preventing the penetration of electrolytes and reducing the oxidation rate of Zn particles [17].

### 3.2. Electrochemical Impedance Spectroscopy Measurement

EIS measurements were used to evaluate the corrosion resistance of coatings with different flake ZnAl pigment contents. The Nyquist and Bode plots of silicate coatings with various pigments immersed in NaCl solution (3.5%, *w/w*) for 60 days are illustrated in Figure 2.

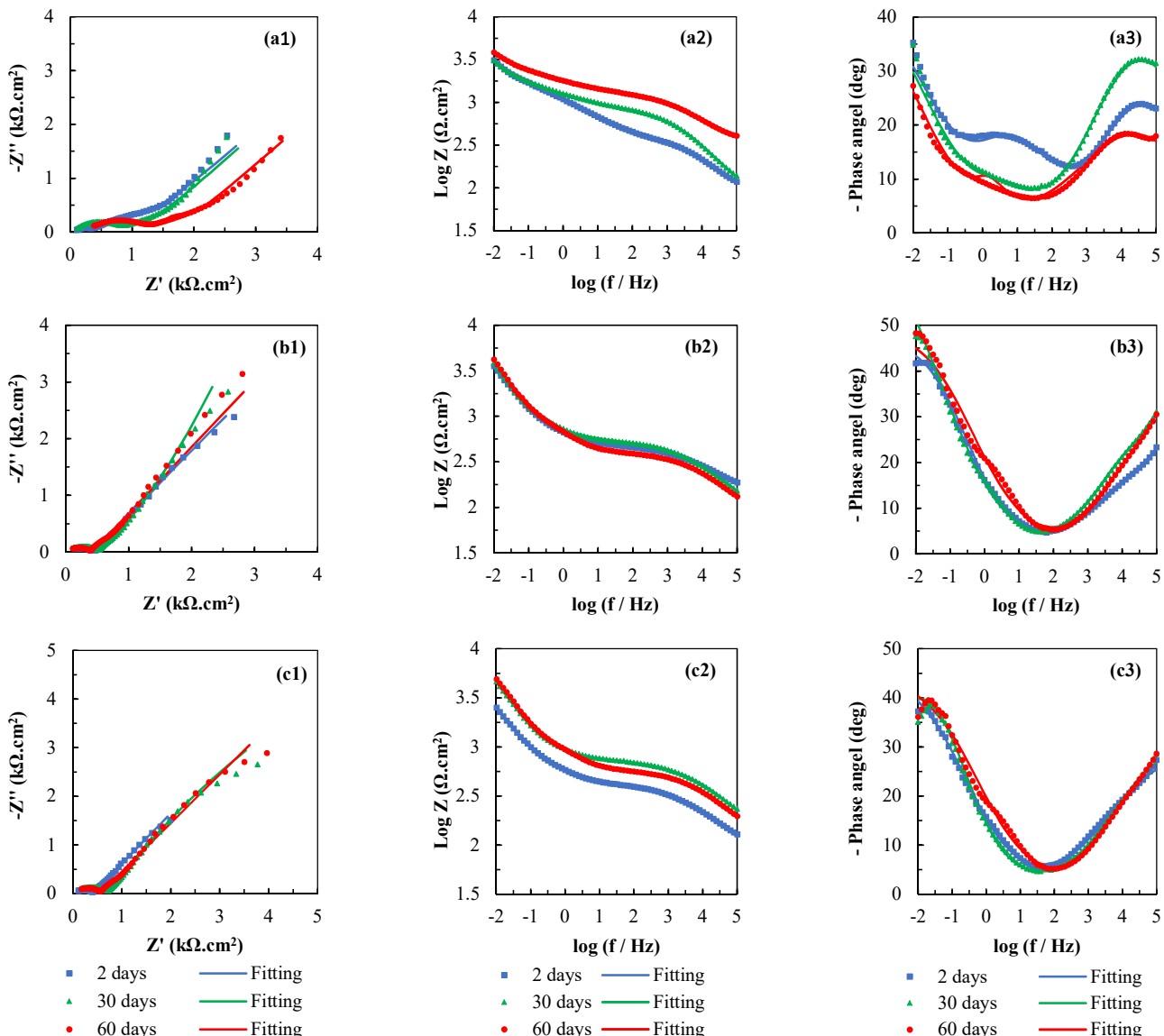

**Figure 2.** Nyquist (series 1) and Bode (series 2 and 3) plots with different pigments of silicate coating (series (**a**): FZnAl20; series (**b**): FZnAl25; series (**c**): FZnAl30) at various immersion times of 2 days, 30 days, and 60 days in 3.5 wt% NaCl.

The Nyquist plots of the three samples containing flake ZnAl pigment (FZnAl20, FZnAl25, and FZnAl30) appear as straight lines with a slope of 45° in the low-frequency range (Figure 2a–c). This result shows that the samples with flake ZnAl pigment coatings have increasing amounts of corrosion products diffused to the microporous channel of the coating after 2 days of soaking. Therefore, Warburg impedance can be used to describe the diffusion process that can prevent electrolyte penetration by corrosion products [23]. The results also show that the impedance of the coatings containing the flake ZnAl alloy pigments increases in the low-frequency region with a long soak time. This result is likely due to both the barrier protection effect and the electrochemical activity of Al [24]. Moreover, the samples with FZnAl20 and FZnAl25 coatings have lower impedance values than the FZnAl30 coatings, indicating that a low concentration of flake ZnAl alloy pigment can reduce the electrolyte diffusion to the substrate. Thus, the oxidation rate of Zn particles is significantly reduced and leads to a decrease in the quantity of corrosion products in the pores of the coating [25].

The EIS data are fitted by the equivalent circuit model, as illustrated in Figure 3. The parameters of the ESI measurements ($R_s$, $R_c$, $R_{ct}$, $C_c$, and $C_{dl}$) allow the electrochemical

properties of the samples to be determined. Double-layer capacitance ($C_{dl}$) and charge transfer resistance ($R_{ct}$) were used to characterize the solubility of the pigments. The Warburg impedance W was used to describe the diffusion process that prevents electrolyte penetration by corrosion products. A constant phase element (CPE) was used as an ideal capacitor to evaluate the influence of the pigments on the roughness of the coatings due to the random distribution of the pigments and inhomogeneity in composition and structure [26]. The total resistance of the silicate coatings was determined via the impedance modulus against immersion time at low frequency (10 mHz), which facilitates understanding of the corrosion behaviour of the silicate coatings [27]. As can be seen in Figure 4a, the samples coated with FZnAl20 and FZnAl25 pigments have lower impedance than the FZnAl30 sample after immersion in an electrolyte solution for 60 days. The increased impedance of the sample with the FZnAl30 coating may be due to a barrier effect of the corrosion products in the pores of the coating [24]. Thus, the flake ZnAl pigment increases the effectiveness of anticorrosion protection and achieves the best protection performance with a flake ZnAl pigment content of less than 30 wt% in the silicate coating.

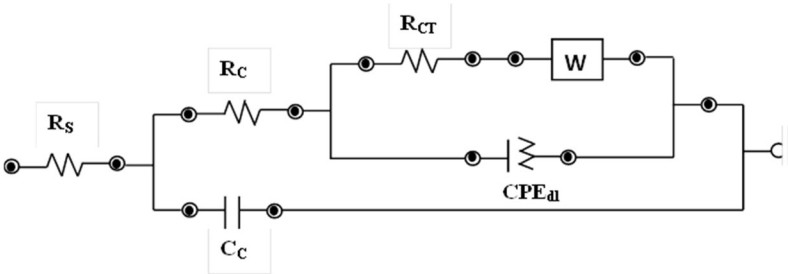

**Figure 3.** The equivalent electric circuit of silicate coatings with flake ZnAl pigments for various immersion times in NaCl (3.5 wt%): $R_s$, $R_c$, $R_{ct}$, $C_c$, $C_{dl}$, and CPE represent solution resistance, coating resistance, charge transfer resistance, coating capacitance, double-layer capacitance, and constant phase element, respectively.

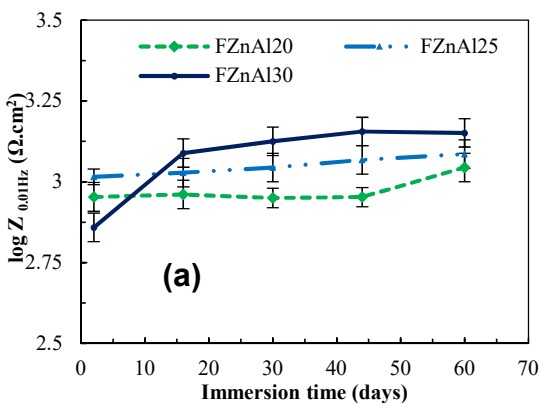
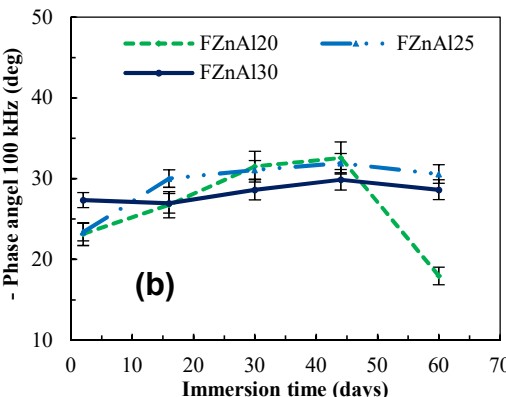

**Figure 4.** The plots of total impedance at low frequency (10 mHz) (**a**) and phase angle at high frequency (100 kHz) (**b**) versus immersion times for the silicate coatings, derived from the Bode plot.

Figure 4b shows the plots of the phase angle at high frequency (100 kHz) versus immersion time derived from the Bode plots, indicating the protective performance of the coatings. The result shows that the phase angle of the samples using FZnAl25 pigments is higher (shifted to the negative side) than the sample using FZnAl20 and FZnAl30 up to 60 days of immersion. This result indicates the high resistance to the electrolyte diffusion into the coating containing FZnAl25 pigments in comparison with the others [24]. However, these samples, especially sample FZnAl20, show gradually decreasing phase angles (shifting to the positive side) after 42 days immersed in the electrolyte solution. This result indicates that the electrolyte penetrates through the coating and reaches the interface between the coating and the steel. In other words, the barrier mechanism of the

coatings using the flake ZnAl pigments is degraded after 42 days of immersion in the electrolyte solution.

### 3.3. Salt Spray Test

The anticorrosive properties of the silicate coatings were evaluated using a salt spray test. The surface images after 1000 h of salt spray test are shown in Figure 5. No white rust appeared on the scratches after 360 h of the salt spray test, suggesting that the flake ZnAl pigment protects against corrosion via a barrier mechanism. However, red rust appeared on the scratch of the samples at different times, with the appearance of red rust on the sample with FZnAl25 pigment taking longer (720 h) than FZnAl20 (72 h) and FZnAl30 (24 h). In other words, the sample FZnAl25 containing the appropriate amount of flake ZnAl pigment reacts favourably with the binder to form a bonding matrix in the silicate coatings [28] and reduces electrolyte diffusion towards the interface between the coating and the substrate. Moreover, there was more red rust on the FZnAL30 sample than on the other samples, indicating that excessive ZnAl content causes a reduction in bond formation in the silicate coatings to create subtle holes and cracks in the coatings [28]. This is likely because a great amount of the silicate binder is consumed, leading to a lack of binding ability of the coatings [14]. As a sequence, the ZnAl pigments are arranged unevenly in the coating matrix, limiting the barrier properties of the flake ZnAl pigments; this excessive content degrades the barrier protection mechanism and increases the diffusion of the electrolyte, leading to reduced anticorrosion performance.

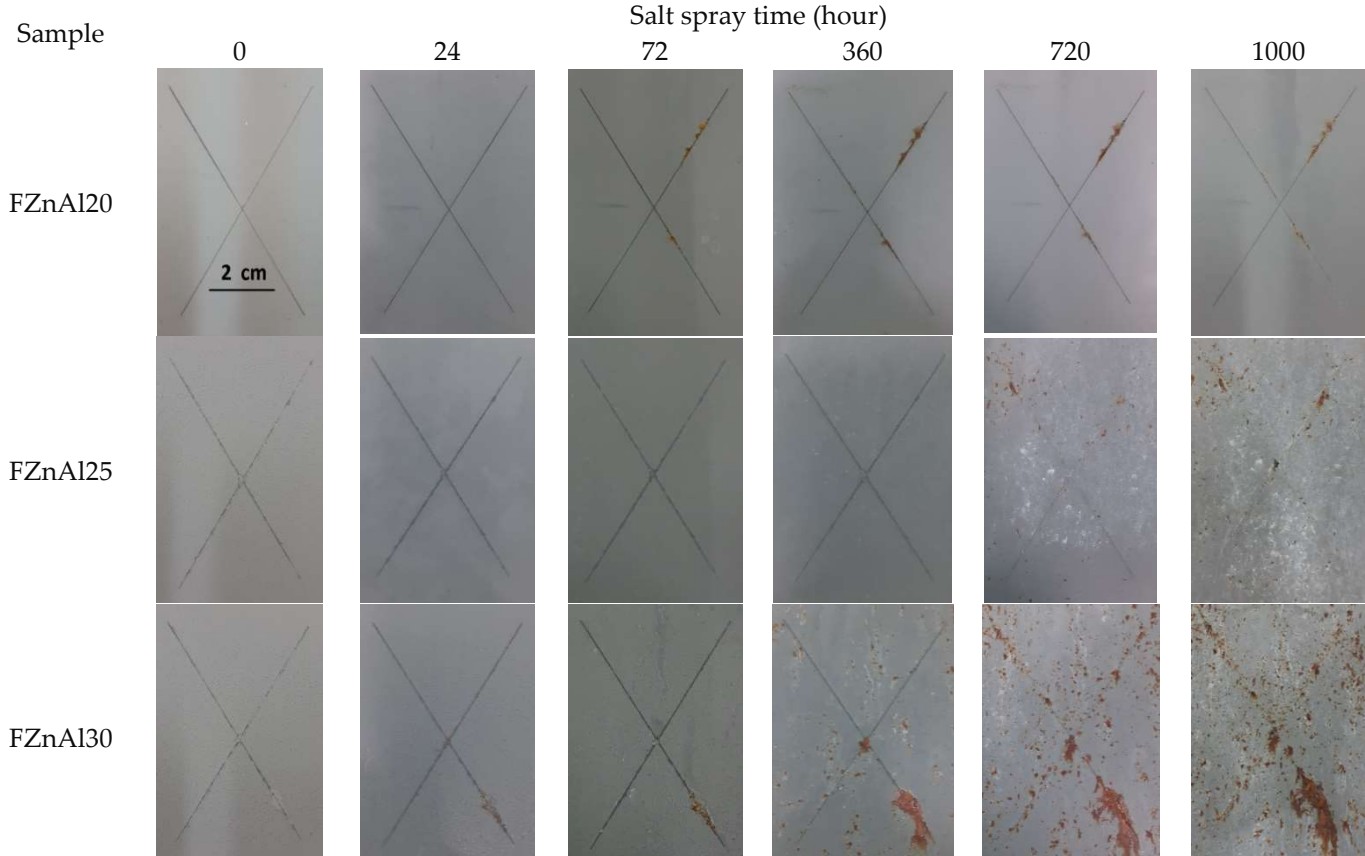

**Figure 5.** Surface images of silicate coatings after exposure to the salt spray testing for 1000 h.

### 3.4. Surface Morphology of Silicate Coating

The surface morphology of the silicate coatings was investigated using SEM measurement before and after the salt spray testing (1000 h), as shown in Figure 6. The result indicates that the coating surfaces provide corrosion barrier protection, produced by the

overlap arrangement of the flake-shaped ZnAl alloy. After the salt spray test, sample FZnAl25 had a more compacted surface, indicating that an appropriate ratio of flake ZnAl pigment to silicate binder can lead to the formation of an effective coating to prevent the penetration of electrolytes into the coating cavity, thereby prolonging the corrosion protection time.

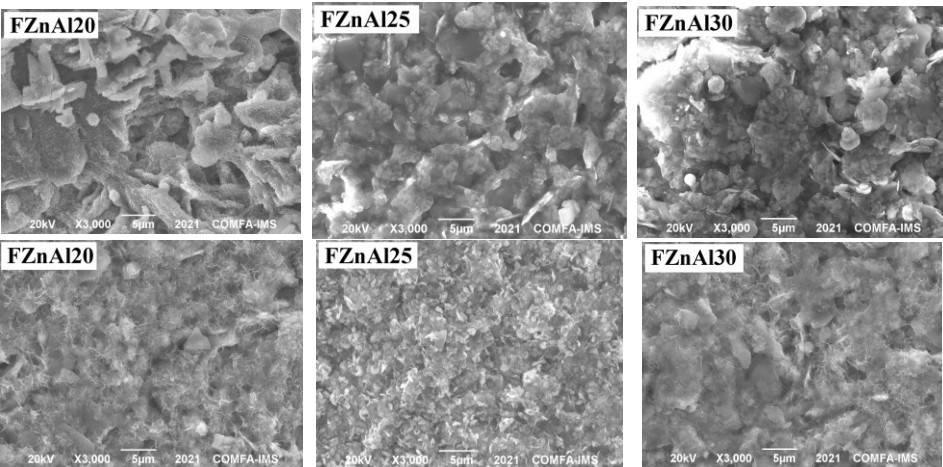

**Figure 6.** SEM images of the silicate coatings with different pigments: before (upper) and after (lower) 1000 h of salt spray test.

### 3.5. Pull-Off Adhesion Tests

The surface images of the silicate coatings and the adhesion strength of the silicate coatings before and after 1000 h of the salt spray are shown in Figure 7. The result shows that all the samples coated with the flake ZnAl alloy pigment exhibited red rust under the coating surface after the salt spray, although the sample using the FZnAl25 coating had very little red rust in comparison with the samples coated with the FZnAl20 and FZnAl30 pigments. This result is consistent with the adhesion loss of the corresponding samples, where the FZnAl25 sample had the lowest degradation of adhesion (37.95%). This result demonstrates that a flake ZnAl alloy content of 25 wt% in the silicate coating forms an effective coating that prevents penetration of the electrolyte into the interface between the coating and the substrate.

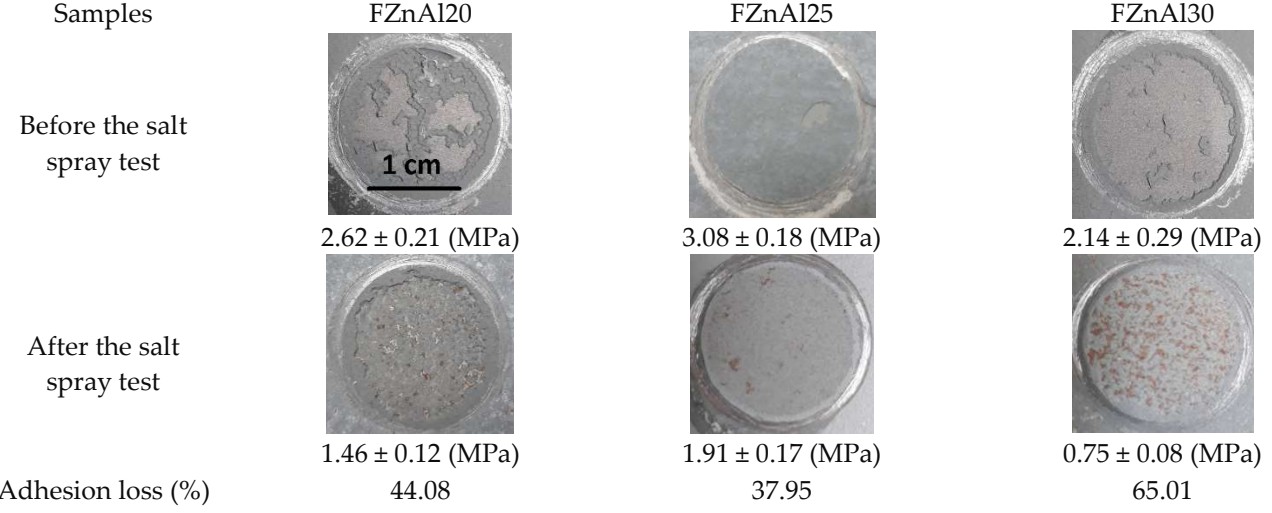

**Figure 7.** Surface images and adhesion of silicate coatings before and after the salt spray test for 1000 h.

## 4. Conclusions

The influence of flake ZnAl alloy concentration in coatings was investigated. Excessive content of ZnAl alloy in the silicate coating causes a decrease in the bonding formation in the coatings and reduces their corrosion protection performance. The salt spray test showed that the time to appearance of rust on the scratch surface in the sample with flake ZnAl 25 wt% in silicate coating was longer than that of the others. The use of a suitable content of flake ZnAl alloy (25 wt%) not only improves corrosion protection but also prevents corrosion products from forming in the silicate coating via a barrier protection mechanism compared to 20 wt% and 30 wt% flake ZnAl alloys. Thus, this work demonstrated good corrosion protection performance of flake ZnAl alloy pigments in the silicate coatings of steel. The advantages of flake ZnAl alloy pigments in water-based inorganic silicate coatings may inspire us to seek and design new materials to achieve coatings with better corrosion protection properties.

**Author Contributions:** Conceptualization, N.H.; methodology, N.H.; validation, T.A.K., L.T.N. and P.M.P.; formal analysis, P.M.P. and T.D.B.; Advisory, T.T.X.H.; project administration, N.H.; funding acquisition, N.H.; visualization, N.V.C.; writing—original draft preparation, N.H.; Writing—reviewing and editing, N.H. and T.-D.N. All authors have read and agreed to the published version of the manuscript.

**Funding:** This research was funded by Vietnam Academy of Science and Technology, Grant No. VAST03.02/21-22.

**Institutional Review Board Statement:** Not applicable.

**Informed Consent Statement:** Not applicable.

**Data Availability Statement:** The authors confirm that the data supporting the findings of this study are available within the article.

**Conflicts of Interest:** The authors declare no conflict of interest.

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
