# Peer review of "Flake ZnAl Alloy as an Effective Pigment in Silicate Coatings for the Corrosion Protection of Steel"

_coatings, doi:10.3390/coatings12081046_

Round 1

Reviewer 1 Report

In this work, the influence of flake ZnAl alloy in silicate coatings on the corrosion protection properties of steel substrates was investigated. The results show that silicate coating containing flake ZnAl alloy (25 wt%) possess the highest total resistance (1417 Ω) and the longest time to the appearance of white rust (720 h). The research aims to develop water-based Zn-rich coatings for environmentally friendly paints, therefore it is significant. But the following comments should be addressed.

(1)   The influence of the shape and content of the ZnAl alloy on the corrosion protection properties of silicate coatings on steel substrates tried to be evaluated in the manuscript. I think that more shapes and particle size distribution of flake ZnAl should be considered in the work.

(2)   The corrosion rate of silicate coating, electrochemical impedance spectroscopy measurement, and pull-off adhesion tests results were similar between FZnAl20 and FZnAl25. The suitable content of flake ZnAl alloy should be reconsidered in the conclusions.

(3)            The influence mechanism of shape and content of the ZnAl alloy on the corrosion protection properties of silicate coatings on steel substrates should be added and discussed.

(4)   “Where βa and βa are Tafel constants from a Tafel Curve and volts/decade of current” in line 96 should be revised.

(5)   The variable symbols should be added in equation (2).

(6)   The title of x-coordinate is invisible in Fig. 2 a1-a3.

(7)   No scale is shown in Fig. 7.

(8)   Please asking an English-native expert in this field to improve the writing. The writing need to have significant improvement.

Author Response

Ms. Ref. No.:  coatings-1826695

Title: Flake ZnAl alloy as an effective pigment in silicate coatings for the corrosion protection of steel.

Journal: Coatings (ISSN 2079-6412)

Dear Editors and Reviewers,

We are grateful to your kind comments. The comments will significantly improve our manuscript. Most of the comments are acceptable and replies to the reviewers are provided as below. The corrected text was highlighted in RED colour in the revised version.

Reviewer #1

Comment: In this work, the influence of flake ZnAl alloy in silicate coatings on the corrosion protection properties of steel substrates was investigated. The results show that silicate coating containing flake ZnAl alloy (25 wt%) possess the highest total resistance (1417 Ω) and the longest time to the appearance of white rust (720 h). The research aims to develop water-based Zn-rich coatings for environmentally friendly paints, therefore it is significant.

Response: Thank you for your careful assessment on our manuscript. 

Comment 1. The influence of the shape and content of the ZnAl alloy on the corrosion protection properties of silicate coatings on steel substrates tried to be evaluated in the manuscript. I think that more shapes and particle size distribution of flake ZnAl should be considered in the work.

Response: Thank you for your comment. We believe that shapes and size of ZnAl are particularly important to get an insight of pigment effect in the coatings. It is an important topic which we are working for the next publications. Aim of this work is to investigate on effect of commercial flake ZnAl pigment content with same size (5-7 μm) that help us get some initial evaluation of flake ZnAl pigment for the anticorrosion properties.

Comment 2. The corrosion rate of silicate coating, electrochemical impedance spectroscopy measurement, and pull-off adhesion tests results were similar between FZnAl20 and FZnAl25. The suitable content of flake ZnAl alloy should be reconsidered in the conclusions.

Response: Thank you for your requirement. The parameters including corrosion rate of silicate coating, electrochemical impedance spectroscopy measurement, and pull-off adhesion tests results were not very similar between FZnAl20 and FZnAl25. However, to make it sense we explained more in the conclusion section.  

Comment 3. The influence mechanism of shape and content of the ZnAl alloy on the corrosion protection properties of silicate coatings on steel substrates should be added and discussed.

Response: Thank you for your requirement. Effect of the alloy content has been discussed in the discussion section. However, the protection mechanism of ZnAl shape will be discussed in the next publications.

Comment 4. “Where βa and βa are Tafel constants from a Tafel Curve and volts/decade of current” in line 96 should be revised.

Response: Thank you for your requirement. It is corrected as follows: “Where βa and βc are anodic and cathodic Tafel constants (V/decade), respectively.”

Comment 5. The variable symbols should be added in equation (2).

Response: Thank you for your requirement. Explains are added to equation (2).

Comment 6. The title of x-coordinate is invisible in Fig. 2 a1-a3.

Response: Thank you for your requirement. It is corrected.

Comment 7. No scale is shown in Fig. 7

Response: Thank you for your requirement. It is corrected.

Comment 8. Please asking an English-native expert in this field to improve the writing. The writing need to have significant improvement.

Response: Thank you for your requirement. It is already corrected by commercial English company in UK style. We also carefully read it again.

Reviewer 2 Report

I have reviewed the manuscript entitled "Flake ZnAl alloy as an effective pigment in silicate coatings for the corrosion protection of steel". My comments on the manuscript are:

1. The language needs to be improved and many mistakes regarding the organization of the sentences are present in whole manuscript. I would strongly suggest read the manuscript several times to avoid these errors.

2. Introduction is very short and fails to provide the real picture of the literature already available.

3. The results have been well discussed and the expressions of all the characterizations tools are appropriately presented.

4. Conclusion should be expended.

5. The authors should focus on working on highlighting the novelty of the work.

Overall, the authors have done a good work and I would like to recommend publication of the manuscript with minor changes.

Author Response

Ms. Ref. No.:  coatings-1826695

Title: Flake ZnAl alloy as an effective pigment in silicate coatings for the corrosion protection of steel.

Journal: Coatings (ISSN 2079-6412)

Dear Editors and Reviewers,

We are grateful to your kind comments. The comments will significantly improve our manuscript. Most of the comments are acceptable and replies to the reviewers are provided as below. The corrected text was highlighted in RED colour in the revised version.

Comment 1. The language needs to be improved and many mistakes regarding the organization of the sentences are present in whole manuscript. I would strongly suggest read the manuscript several times to avoid these errors.

Response: Thank you for your requirement. It is read and corrected errors.

Comment 2. Introduction is very short and fails to provide the real picture of the literature already available.

Response: Thank you for your kind requirement. We have added more information in the introduction section.

Comment 3. The results have been well discussed and the expressions of all the characterizations tools are appropriately presented.

Response: Thank you for your good assessment on our manuscript.

Comment 4. Conclusion should be expended..

Response: Thank you for your comment. We added more information in the conclusion.

Comment 5. The authors should focus on working on highlighting the novelty of the work.

Response: Thank you for your comment. We added as follows: “The advantages of flake ZnAl alloy pigments in water-based inorganic silicate coatings may inspire us to seek and design new materials to achieve coatings with better corrosion protection properties”

Comment 6. Overall, the authors have done a good work and I would like to recommend publication of the manuscript with minor changes.

Response: Thank you for your recommendation.
